

# Preliminary studies of selected *Lemna* species on the oxygen production potential in relation to some ecological factors

Joanna Sender[1] and Monika Różańska-Boczula[2]

[1] Department of Hydrobiology and Protection of Ecosystems, University of Life Sciences in Lublin, Lublin, Poland
[2] Department of Applied Mathematics and Computer Science, University of Life Sciences in Lublin, Lublin, Poland

## ABSTRACT

Dissolved oxygen is fundamental for chemical and biochemical processes occurring in natural waters and critical for the life of aquatic organisms. Many organisms are responsible for altering organic matter and oxygen transfers across ecosystem or habitat boundaries and, thus, engineering the oxygen balance of the system. Due to such *Lemna* features as small size, simple structure, vegetative reproduction and rapid growth, as well as frequent mass occurrence in the form of thick mats, they make them very effective in oxygenating water. The research was undertaken to assess the impact of various species of duckweed (*L. minor* and *L. trisulca*) on dissolved oxygen content and detritus production in water and the role of ecological factors (light, atmospheric pressure, conductivity, and temperature) in this process. For this purpose, experiments were carried out with combinations of *L. minor* and *L. trisulca*. On this basis, the content of oxygen dissolved in water was determined depending on the growth of duckweed. Linear regression models were developed to assess the dynamics of changes in oxygen content and, consequently, organic matter produced by the *Lemna*. The research showed that the presence of *L. trisulca* causes an increase in dissolved oxygen content in water. It was also shown that an increase in atmospheric pressure had a positive effect on the ability of duckweed to produce oxygen, regardless of its type. The negative correlation between conductivity and water oxygenation, obtained in conditions of limited light access, allows us to assume that higher water conductivity limits oxygen production by all combinations of duckweeds when the light supply is low. Based on the developed models, it was shown that the highest increase in organic matter would be observed in the case of mixed duckweed and the lowest in the presence of the *L. minor* species, regardless of light conditions. Moreover, it was shown that pleustophytes have different heat capacities, and *L. trisulca* has the highest ability to accumulate heat in water for the tested duckweed combinations. The provided knowledge may help determine the good habitat conditions of duckweed, indicating its role in purifying water reservoirs as an effect of producing organic matter and shaping oxygen conditions with the participation of various *Lemna* species.

Corresponding author
Monika Różańska-Boczula,
monika.boczula@up.lublin.pl

## INTRODUCTION

Dissolved oxygen (DO) in water is fundamental for chemical and biochemical processes occurring in natural waters and critical for the life of aquatic organisms (*Wetzel, 2001*). Oxygen is consumed in various biological and biogeochemical processes, mainly during organisms' respiration and decomposition (degradation and mineralisation) of dead organic matter. Oxygen dynamics in lakes are driven by internal lake processes and gas exchange with the atmosphere. It is widely accepted that the transfer of $O_2$ across the water surface occurs by diffusion (*Dugan et al., 2016*). The concentration of this gas in water and sediment depends on the amount of exchange with the atmosphere, water temperature, and rate of decomposition of organic matter, as well as the intensity of photosynthesis and respiration. Because oxygen shows relatively low solubility in water (*Broecker & Peng, 1982*), with its diffusion several orders of magnitude slower than in air (*Sculthorpe, 1967*), organisms in aquatic systems can influence oxygen conditions in long and shorter time scales (*Miranda, Driscoll & Allen, 2000*; *Caraco & Cole, 2002*; *Caraco et al., 2006*). The availability of oxygen in aquatic environments is vital for the survival of many organisms, and the balance of oxygen production and consumption is a key factor in maintaining a healthy ecosystem.

Aquatic green plants and some bacteria absorb light energy, which is further used to synthesise carbohydrates from inorganic compounds ($CO_2$, $H_2O$) by releasing gaseous oxygen. Duckweeds are some of the most miniature herbaceous plants, ranging from 1 to 5 mm in length (only *L. trisulca* reaches a size of 6 to 10 mm). Plants with a body simplified to the organisation of thallus plants, whose only distinguishable organs are reduced flowers and roots. They are annual plants that often grow in still or slow-flowing shallow eutrophic waters (*Landolt, 1986*; *Cross, 2005*; *Bog et al., 2020*). Duckweeds are monocotyledonous macrophytes belonging to the Lemnaceae family, and classified as hydrophytes, free-floating on the water surface, with survival organelles wholly submerged in water and sinking to the bottom during unfavourable seasons (*Movafeghi et al., 2013*). They are widespread and comprise 37 different species globally (*Chakrabarti et al., 2018*). In Poland there are six duckweed species, and all of them belong to the native flora of Poland. The reasons for the abundance of *L. minor* and *L. trisulca* in Polish waters are their ecological plasticity, short life cycle, and rapid reproduction. In addition, the lack of natural competitors and climatic conditions favourable to their development and their simplicity of dispersal. Their luxuriant bloom can cause the development of a thick green mat that covers up to 100 per cent of the water surface, cutting off the access of light to the water depths and often causing submerged plants to recede (*Smolders, Lucassen & Roelofs, 2022*; *Sender, 2012*; *Sender et al., 2021*). Under natural conditions, duckweeds are a valuable food source for many herbivorous organisms, including water birds and fish (*Drost, Matzke & Backhaus, 2007*). *L. minor* is used as food in several Asian countries because of sufficient

amounts of protein, starch and fatty acids (*Mwale & Gwaze, 2013*; *Ibrahim et al., 2017*). Other than human food, *L. minor* is also used as livestock and poultry feed and has been reported to be very nutritious, providing phosphates and proteins (*Li et al., 2020*; *Zhao et al., 2012*; *Chepkirui et al., 2022*; *Chepkirui et al., 2023*).

*Lemna* mats create unique floating habitats that significantly increase biodiversity, leading to greater complexity of the food web and an increase in the overall production of organic matter in the ecosystem (*Mitsch & Gosselink, 2015*). *Lemna* is an excellent plant for improving water quality, among others, due to its ability to absorb harmful substances present in water. It accumulates heavy metals (*Ali et al., 2016*; *Iannelli et al., 2022*; *Sekomo et al., 2012*), municipal pollutants, moreover absorb nitrogen (*Singh, Misra & Pandey, 2008*; *Devlamynck et al., 2020*) and phosphorus compounds (*Ceschin et al., 2019a*; *Ceschin, Crescenzi & Iannelli, 2020a*; *Paterson, Camargo-Valero & Baker, 2020*), antibiotics (*Baciak et al., 2016*) and even works as a bioindicator of microplastic pollution (*Rozman & Kalčíková, 2022*). Numerous works (*Axtell, Sternberg & Claussen, 2003*; *Oporto et al., 2006*; *Frédéric et al., 2006*; *Drost, Matzke & Backhaus, 2007*; *Alvarado et al., 2008*) also confirm the potential for purifying water bodies through binding selected elements in duckweed biomass. A distinction is made between biosorption (dead biomass) and bioaccumulation (by living matter) (*Chojnacka, 2006*). The importance of duckweeds in water bodies is determined primarily by its ubiquity in all climate zones, except deserts and possibly tundra (*Drost, Matzke & Backhaus, 2007*; *Axtell, Sternberg & Claussen, 2003*) rapid growth and ease of culture (*Drost, Matzke & Backhaus, 2007*; *Ceschin et al., 2016*; *Chakrabarti et al., 2018*; *Chepkirui et al., 2023*); wide range of tolerance to changes in temperature and pollutants. Since duckweeds grow fast and are easy to keep in culture, they have been widely used in physiological (*Prasad et al., 2001*; *Hou et al., 2007*; *Zezulka et al., 2013*), as well as eco-toxicology studies (*Radić et al., 2010*; *De Alkimin et al., 2020*). Nevertheless, their mass presence seriously reduces the amount of light in deeper layers of water, which limits the photosynthesis among submerged plants (*Cataneo et al., 1998*; *Ceschin et al., 2020b*). Finally, thick beds of floating-leaved macrophytes can prevent gas exchange and make low DO levels in the water column more severe (*Caraco & Cole, 2002*; *Bradshaw, Allen & Netherland, 2015*). There are many studies on the contribution of particular groups of macrophytes in gas exchange: emergent (*Bunch, Allen & Gwinn, 2010*), submerged (*Frodge, Thomas & Pauley, 1990*), floating-leaved (*Caraco & Cole, 2002*), or mixed submerged and emergent (*Miranda & Hodges, 2000*), all of which have shown improved DO concentrations in the water. Different *Lemna* species display different characteristics and play different roles in the dynamics of organic matter and oxygen. Both *L. minor* and *L. trisulca* are the most common duckweeds in water ecosystems. However, most of the cited literature covers research conducted on various species of duckweeds, including *L. minor*, and often excluding *L. trisulca*. The impact of *Lemna*, especially *L. minor*, on the oxygen content in water was studied, among others, by *Ceschin et al. (2019b)*, *Ceschin, Crescenzi & Iannelli (2020a)* and *Ceschin et al. (2020b)*, emphasising the strong impact of floating duckweed mat on the oxygen content in water and, consequently, on the survival of not only plant but also animal communities.

Quite surprisingly, there is little work on the participation of two species (*L. minor* and *L. trisulca*) together or separately in water oxygenation. To fill this gap, a study was conducted to assess the effect of two different duckweed species on the dissolved oxygen content in water as well as detritus formation. Based on a preliminary study, we hypothesise that the quantity of oxygen and detritus (organic matter) varies between the two *Lemna* species. Another research problem was determining the role of ecological factors such as light, atmospheric pressure, temperature, and conductivity in oxygenation. It allowed for determining predictive models for oxygen content and ecological factors for selected duckweed species.

## MATERIALS & METHODS

Biological material in the form of duckweeds was obtained, using a mill gauze scoop, from two oxbow lakes located in the Nadwieprzański Landscape Park (N51°17′03.38″, E22°52′19.11″ and N51°15′57.51″, E22°56′21.60″). They represented eutrophic reservoirs with average parameter values: pH 7.2, visibility 0.6 m, conductivity 380 $\mu$S m$^{-3}$. The analyses were performed *in situ* using the MultiLine P4 multi-parameter meter, as well as TP 0.05 mgP dm$^{-3}$, 1.0 mgN dm$^{-3}$ and Chla 10.1$\mu$g dm$^{-3}$, as well as measurements made in the laboratory using the ICS 5000 Ion Chromatograph. The oxbow lakes have surface areas from 0.5 to 1.5 ha and are primarily surrounded by grasslands. All species occurred in the same habitat, covering 80% of the oxbow lakes with a slight dominance of *L. minor* (65%). Two experimental protocols were used in this study. The material was sterilised after being brought to the laboratory (*Bowker, Duffield & Denny, 1980*). After sterilisation, the plants were washed with distilled water. We preincubated the plants, 50 fonds of each species in 0.5 l glass beakers a culture Steinberg's medium in homogeneous conditions, under full light (light was provided by metal halide bulbs (Osram 250 W) at a photon flux density of 300 $\mu$mol m$^{-2}$ s$^{-1}$) and constant temperature conditions of 20 °C ±2. Uncontaminated duckweed fronds were acclimated and allowed to vegetatively multiply for two months. Preliminary analyses were carried out including O2 (%), O2 (mg/l), water temp (°C), ambient temperature (°C), light (lux), coverage (%), thickness of *Lemna* layer (mm), sediment cover (%), sediment height (cm), in cultures of two species of duckweed. The apparatus used was the same as in the experiment. The Mann-Witney U test was used to compare the results between the two species (*L. minor* and *L. trisulca*). According to the concept of an experiment, the multiplied duckweed material was transferred to beakers filled to 500 ml with distilled water. Twenty glass beakers were prepared: 6 with *L. minor*, 6 with *L. trisulca*, 6 mixed, and 2 controls. Ten fronds (n = 10) for each species were transferred into a glass beaker separately in 3 repetitions (Table 1). The control samples contained only distilled water, without duckweed. Duckweed was grown on Steinberg's growth medium, modified by Altenburger (*Wang, 1990*; *Zhang et al., 2010*; *Vidaković Cifrek, Sorić & Babić, 2013*; *Bergkamp, 2013*). To limit access to light (twilight), the walls of the beaker were shaded (covered with aluminium foil), and the bottom and top of the beaker remained transparent (Table 1). The top of the beaker was sealed with Parafilm to reduce evaporation.
**Table 1  Sample labeling scheme.**

| Species composition | Shaded / limited light | Full light |  |
|---|---|---|---|
| *Lemna trisulca* | 1A | 1B | |
| *Lemna minor* | 2A | 2B | |
| *Lemna minor* and *L. trisulca* | 3A | 3B | |
| Control, no *Lemna* | 4A | 4B | |

The following measurements were taken from the first day after setting up the culture that was placed in the laboratory in a place with a temperature ranging from 18.5 to 28.5 °C and variable lighting conditions (depending on natural light). They were variable and dependent on laboratory conditions. Due to the duration of the experiment, the temperature in the laboratory was subject to significant fluctuations, so it was included as an experimental factor. The research concerned: ambient and water temperature in the beaker (Temperature Logger TG-4100), amount of light reaching the water surface (Luxmeter 540 max), oxygen content in $mgO_2$ $dm^{-3}$, and oxygen saturation of water (%) (multi-parameter meter Hanna HI 98194). Moreover, the degree of *Lemna* coverage on the water surface in the beaker (%), width and thickness (cm) of the layer formed by the plants, and amount of sediment formed by living matter: coverage (%) and thickness (cm) were determined. ImageJ image analysis software was used to assess the duckweed coverage of the beaker, and a ruler was used to measure the thickness of sediments and floating plants. Atmospheric pressure was measured in the laboratory using an electronic barometer Testo 511.

Due to the variability of external conditions (light, temperature, pressure), a fixed measurement time was established, at noon. The duration of the experiment was related to the achievement state when the parameters did not change (following four months). After 21 days, the first signs of vegetative growth inhibition were observed. The heat storage capacity of selected species of duckweed was also determined according to the following formula (*Lange & Navrotsky, 1993*):

$$Q = \sum_{Z=0}^{Z} F_z t_z z$$

Q - heat input (cal)
z - depth (cm)
$F_z$ - surface area ($cm^2$)
$t_z$—water temperature ( °C)

## Statistical analysis

The statistical analysis aimed to determine the effects of two factors, namely the type of duckweed (control, *L. trisulca, L. minor*, mixed duckweed) and varying light conditions (full, shade) on dissolved oxygen content ($O_2$ %, $O_2$ mg/l), conductivity, and the dynamics of duckweed development (coverage, layer height, amount of sediment) under specific thermal and light conditions. The appropriate statistical method was selected by verifying data conformity to a normal distribution (Shapiro–Wilk test). The Gaussian distribution was only confirmed for trait $O_2$ (mg/l) (after discarding two outliers); therefore, Anova two-factor analysis of variance and T-Tukey post-hoc tests were used to verify the significant effect of the duckweed and light combination on the selected trait. In other cases, despite the application of the Bliss transformation, a normal distribution could not be achieved, so traits $O_2$ (%), cover (%), layer thickness, sediment cover (%), sediment height, and conductivity were subjected to non-parametric analysis. The influence of factors was verified using the Kruskal-Wallis test. A post-hoc analysis in the form of multiple comparisons was also performed. Statistically significant results were visualised in graphs where letter designations indicated homogeneous groups.

Spearman's rank correlation was used to determine the strength and direction of the relationship between $O_2$ (mg/l), $O2$ (%), conductivity, water and ambient temperature, light, and atmospheric pressure. The significance of the correlations provided the basis for the development of regression models describing the change in dissolved oxygen as a function of atmospheric pressure and conductivity for full light and shade, respectively. The significance of the regression models was verified with an F test (Fisher-Snedecor). Moreover, linear trend models were estimated for sediment parameters (cover, thickness). The coefficients of determination ($R^2$) confirmed a very good fit of the models to the experimental data, providing the basis for their use for predictive purposes. The acceptability of the forecast was verified based on the relative error of the prediction. The analysis of the study results was performed using STATISTICA 13.3, assuming a significance level of $\alpha = 0.05$.

## RESULTS

Analyses of the development of the various factors depending on the duckweed species revealed that three characteristics, namely water temperature, ambient temperature, and amount of incoming light, did not differ significantly. For the remaining seven traits, the Mann–Whitney U test showed statistically significant differences ($p < 0.05$, Table 2).

The average oxygen saturation (over 128%, 10.6 mg/l) in *L. trisulca* samples was 30% higher, and the water temperature was also slightly higher (by 1%) than for *L. minor*. *Lemna minor* occupied a significantly larger area of the water surface in the backer (70% more). The thickness of the *Lemna* layer was almost 2.5 times higher for *L. trisulca*. In the case of sediment cover, *L. trisulca* occupied on average 1.5% more of the vessel surface than *L. minor*, and the sediment height for *L. trisulca* was 1.5 times higher (Table 3). The differences in oxygen content for the different species were also significant. Higher oxygen supersaturation of water was frequently observed in *L. trisulca* (Table 3). The trait with the
**Table 2 Mann-Whitney U test –comparison of characteristics between *L. minor* and *L. trisulca*.**

| Feature | U test | *p*-value |
|---|---|---|
| $O_2$ (%) | 160 | 0.0012* |
| $O_2$ (mg/l) | 176 | 0.0031* |
| Water temperature (C°) | 304 | 0.5396 |
| Ambient temperature (C°) | 330 | 0.8906 |
| Light (lux) | 280 | 0.2924 |
| Conductivity (µS/cm) | 1160 | 0.0011* |
| Coverage of *Lemna* layer (%) | 0 | 0.0000* |
| Thickness of *Lemna* layer (mm) | 36 | 0.0000* |
| Sediment cover (%) | 244 | 0.0204* |
| Sediment height (cm) | 142 | 0.0003* |

**Notes.**
*$p < 0.05$

**Table 3 Descriptive statistics of the studied features (V (%)—coefficient of variation).**

| Species | *Lemna trisulca* | | | *Lemna minor* | | |
|---|---|---|---|---|---|---|
| Feature | Mean | stat. dev. | V | Mean | stat. dev. | V |
| $O_2$ (%) | 128.34 | 33.59 | 26 | 98.82 | 22.71 | 23 |
| $O_2$ (mg) | 10.55 | 2.32 | 22 | 13.10 | 17.03 | 130 |
| Water temperature | 24.83 | 2.88 | 12 | 24.60 | 2.99 | 12 |
| Ambient temperature | 23.53 | 2.62 | 11 | 23.48 | 2.51 | 11 |
| Light | 705.92 | 398.12 | 56 | 692.23 | 645.37 | 93 |
| Conductivity | 670.85 | 76.55 | 11 | 617.46 | 79.23 | 13 |
| Coverage of *Lemna* layer | 17.38 | 6.96 | 40 | 88.08 | 10.56 | 12 |
| Thickness of *Lemna* layer | 38.15 | 13.63 | 36 | 14.62 | 5.00 | 34 |
| Sediment cover | 96.92 | 10.87 | 11 | 95.23 | 7.26 | 8 |
| Sediment height | 0.61 | 0.14 | 23 | 0.41 | 0.18 | 43 |

highest but moderate variability for *L. trisulca* was light (variation coefficient V = 56%), while for *L. minor* it was $O_2$ content (V = 130%, high variability).

Further observations were made for the features shown in Table 3. This time, four combinations of duckweeds (*L. trisulca*, *L. minor*, mixed, control) with different light access (partial shade, full light) were studied experimentally, and features similar to those recorded in the pilot study were observed.

## Oxygen concentration in individual duckweeds

The Kruskal-Wallis test showed a significant effect of the combination of duckweed and the amount of light on $O_2$ concentrations in water (Table 4). Similarly, the interaction effect was found to be significant ($p = 0.0121$), meaning that the impact of one factor (type of duckweed) altered the effect of the other factor (quantity of light). As indicated in Fig. 1, the saturation of water with oxygen was the highest in mixed duckweed in half shade (86%, 7.13 mg/l), and the lowest in control water in half shade (69.62%, 7.77 mg/l). The multiple comparisons test showed a significant difference between the $O_2$ percentage

**Table 4** The influence of duckweed species and light on the studied features: H, result of Kruskal-Wallis test; F, result of Anova test; df, degrees of freedom.

| Sources of variation | | Duckweed variant | Light | Interaction |
|---|---|---|---|---|
| O$_2$ (%) | H | 9.61 | 4.46 | 17.97 |
| | df | 3 | 1 | 7 |
| | p-value | 0.0222* | 0.0347* | 0.0121* |
| O$_2$ (mg) | F | 1.48 | 4.64 | 0.37 |
| | df | 3 | 1 | 3 |
| | p-value | 0.2274 | 0.0343* | 0.7713 |
| Coverage (%) | H | 39.98 | 0.06 | 40.54 |
| | df | 2 | 1 | 5 |
| | p-value | <0.0001* | 0.8058 | <0.0001* |
| Thickness of *Lemna* layer (mm) | H | 17.19 | 0.8 | 18.42 |
| | df | 2 | 1 | 5 |
| | p-value | 0.0002* | 0.3721 | 0.0025* |
| Sediment cover (%) | H | 39.98 | 0.06 | 40.54 |
| | df | 2 | 1 | 5 |
| | p-value | 0.0001* | 0.8058 | 0.0001* |
| Sediment thickness (cm) | H | 11.19 | 0.27 | 13.24 |
| | df | 2 | 1 | 5 |
| | p-value | 0.0037* | 0.6052 | 0.0213* |

**Notes.**
*$p < 0.05$

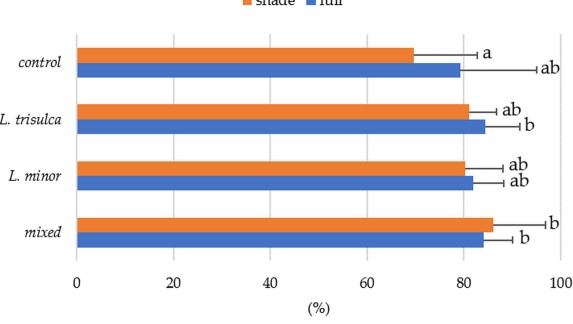

**Figure 1** Average percentage oxygen saturation of water as a function of water lash variant and light access (horizontal segment indicates standard deviation; same letters indicate homogeneous groups).

determined for control water in semi-shade and mixed and three-row duckweed in full light. No statistically significant differences were found for the other combinations (Fig. 1).

The mean value of water oxygen saturation in various duckweed systems varied (Fig. 1). In full light, the highest values occurred in samples with *L. trisulca* (average 84.42%, 7.0 mg/l), while in shaded samples the highest values occurred for mixed duckweed (average 86%, 7.13 mg/l). Oxygen content varied significantly only according to light access (Table 4, $p = 0.0343$). This result was confirmed by a post-hoc T-Tukey test (average O$_2$ values in
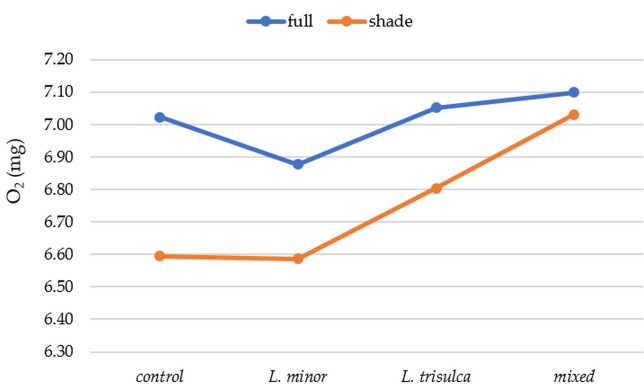

**Figure 2  Average O₂ depending on duckweed variant and light access.**

full light are higher than in partial shade, $p = 0.0379$). Regardless of the duckweed species, higher $O_2$ values in water were recorded under full light (Fig. 2). Higher $O_2$ contents were recorded in the control sample than in samples from *L. minor*.

### Lemna coverage

Coverage was found to vary significantly between duckweed, while light intensity was insignificant for this feature (Table 4). The results for mixed duckweed are distinguishable from the other two (Fig. 3, letter designations). Post-hoc tests of multiple comparisons ($p < 0.0001$) confirmed this relationship. Irrespective of light access, significantly higher coverage occurred in mixed duckweed, reaching between 14% and 15%. The least area was occupied by *L. trisulca*, from 3.6 to 4%. These values were slightly lower in samples with full light access (Fig. 3). This is related to the ecology of this species.

### Thickness of *Lemna* layer

Due to the varying thickness of the plant layer formed, particularly in the case of *L. trisulca*, the effect of the amount of light and the type of duckweed on the thickness of this layer was analysed. The results showed that the thickness of the plant layer varied significantly according to the duckweed species, while the light factor was not statistically significant in this case (Table 4). *L. trisulca* reached its greatest thickness in full light (average 29 mm), which was three times higher than for *L. minor* (average 9.5 mm). Post-hoc tests of multiple comparisons further showed a significant difference between the thickness of the layer of *L. trisulca* and mixed duckweed in full light and that of *L. minor* (Fig. 4).

The analysed duckweed species accumulated different amounts of heat. Larger values of cover and the thickness of the plant mat certainly influenced the amount of heat they retained. *L. trisulca* samples collected more in both full-light exposed and shaded samples, reaching 4866 cal and 4855 cal respectively, while in *L. minor* samples, the accumulated heat was 4838 cal.

### Sediment cover

The dynamics of its accumulation (formation) differed according to the type of duckweed (Table 4). The interaction of the main factors (duckweed type and light) was also significant

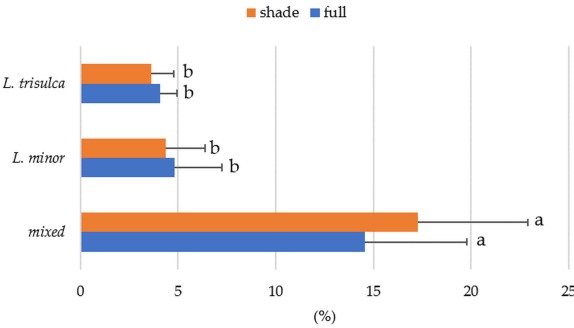

**Figure 3** Average coverage as a function of duckweed variant and light access (horizontal segment indicates standard deviation; the same letters indicate homogeneous groups).

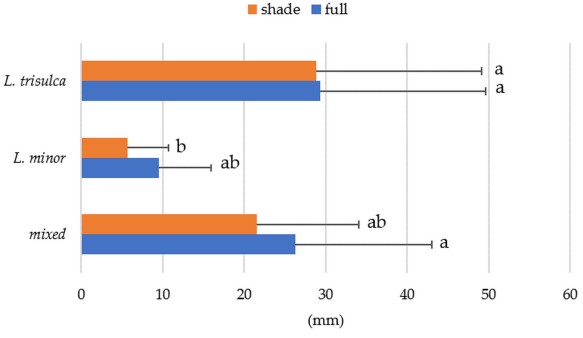

**Figure 4** Average layer thickness as a function of duckweed variant and light access (horizontal segment indicates standard deviation; the same letters indicate homogeneous groups).

for forming this trait. The post-hoc test results of multiple comparisons indicated that the sediment coverage for *L. trisulca* at full light was significantly higher than in the other cases (Fig. 5A).

Similarly to sediment cover, sediment height did not vary significantly with light access (Table 4, $p > 0.05$) but did vary with the combination of the main factors (duckweed type, light), as indicated by the significance of the interaction (Table 4, $p < 0.05$). Sediment thickness was significantly greater for light-limited mixed duckweed than for *L. minor*, regardless of light conditions (Fig. 5B). Sediment from the fallen remains of all duckweed combinations appeared in the first several days (depending on the duckweed combination, it was the second or third day). The coverage did not exceed 20%. The fastest, as early as day 3, coverage was as high as 60% with *L. trisulca*. On day 25, a slower but equally high increase occurred among the duckweed combinations (58%). An evident increase in the amount of sediment occurred in all combinations around day 40. In the case of *L. trisulca* and mixed duckweeds, it continued further, reaching a maximum value of 99% on day 60 of observation. An increasing trend was observed for all duckweed combinations, leading to a differentiated maximum sediment coverage. The daily increase in sediment cover was

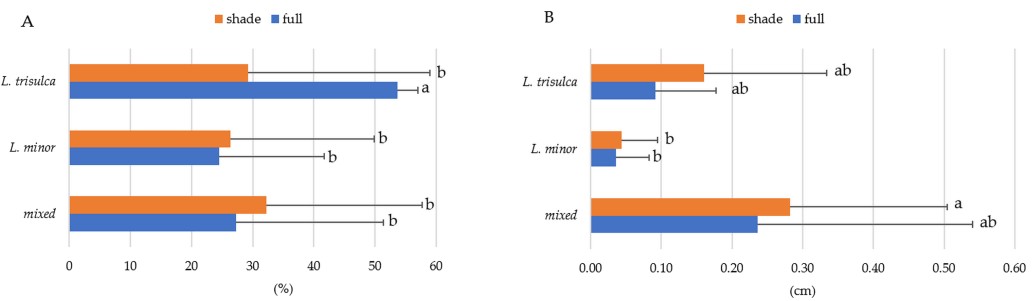

**Figure 5 Average sediment cover (A) and average sediment height (B) according to 'duckweed variant' and light access (the horizontal segment indicates the standard deviation; the same letters indicate homogeneous groups).**

**Table 5 Linear development trend model for sediment ($R^2$, determination coefficient).**

| D. type | Sediment | Regression ($x$ = time in days) | $R^2$ | Trend description |
|---|---|---|---|---|
| 1A | Coverage | $y = 1.482x - 1.261$ | 0.94 | Increasing trend over the whole observation period |
| 1A | Thickness | $y = 0.0107x - 0.0411$ | 0.98 | Increasing trend up to the 38th day of observation then a constant level equal to 0.4 (cm) |
| 1B | Coverage | $y = 0.568x + 44.153$ | 0.59 | The 50% coverage only appears on the 3rd day of observation (previously insignificant). The last day of observation revealed a 60% increase in coverage compared to the 38th day of observation |
| 1B | Thickness | $y = 0.0055x + 0.0021$ | 0.91 | Increasing trend over the whole observation period; between the 38th and 61st day of observation a 4-fold increase in value |
| 2A | Coverage | $y = 1.491x - 1.084$ | 0.98 | Increasing trend up to the 38th day of observation, thereafter constant level equal 55% |
| 2B | Coverage | $y = 0.803x + 8.051$ | 0.84 | Increasing trend from day 2 of observation |
| 3A | Coverage | $y = 1.525x + 4.239$ | 0.83 | Increasing trend up to the 38th day of observation, thereafter constant level equal 60% |
| 3A | Thickness | $y = 0.0117x + 0.046$ | 0.83 | Increasing trend up to the 36th day of observation, then constant level equal 0.6 (cm) |
| 3B | Coverage | $y = 1.222x + 2.830$ | 0.79 | Increasing trend up to the 38th day of observation, thereafter stable level equal 70% |
| 3B | Thickness | $y = 0.0149x - 0.0693$ | 0.91 | Increasing trend over the whole observation period |

**Notes.**

D. type: 1A (L. trisulca/shadow); 1B (L. trisulca/full light); 2A (L. minor/shadow); 2B (L. minor/full light); 3A (L. mixed/shadow); 3B (L. mixed/full light).

most remarkable in mixed duckweed in shadow (average by 1.53% of the beaker area) and lowest in *L. trisulca* in full light (Table 5).

The growth of the sediment layer in thickness occurred most intensively in mixed duckweed with full light. For *L. minor* (2A, 2B), it was insignificant. The most intensive growth occurred between days 30 and 40, particularly in mixed duckweed and *L. trisulca* under full light. The other duckweed arrangements showed no growth after this period (Table 5). The predictive values obtained for the amount of collected sediment under undisturbed conditions in reservoirs with duckweed indicate that the largest increments of approximately 5.3 cm will occur in the case of mixed duckweed and 2 cm for *L. trisulca*

**Table 6  Prognostic values for sediment thickness and sediment cover.**

| Sediment | Projection horizon | 3B | 3A | 1B | 1A | 2B | 2A |
|---|---|---|---|---|---|---|---|
| Sediment thickness (cm) | 24 h | 0.02 | 0.01 | 0.01 | 0.01 | 0 | 0 |
| | 30 days | 0.38 | 0.40 | 0.17 | 0.28 | 0.09 | 0.1 |
| | 365 days | 5.37 | After 1 month const. 0.6 | 2.01 | After 1 month const. 0.4 | After 1 month const. 0.1 | After 1 month const. 0.1 |
| | Relative forecast error | 10%[**] | | 8%[**] | | | |
| Sediment coverage (%) | 24 h | 1.22 | 1.53 | 0.57 | 1.48 | 0.66 | 1.49 |
| | 30 days | 39.49 | 49.99 | 61.19 | 43.20 | 33.43 | 43.65 |
| | 365 days | After 2 months const. 70% | After 2 months const. 60% | After 2 months const. 95% | After 2 months const. 92% | After 2 months const. 50% | After 2 months const. 55% |

**Notes.**
[**]The forecast value is acceptable when the relative forecast error does not exceed 10%; 1A (L. trisulca/shadow); 1B (L. trisulca/full light); 2A (L. minor/shadow); 2B (L. minor/full light); 3A (L. mixed/shadow); 3B (L. mixed/full light); const. means constantly.

per year (Table 6). The largest daily and annual increments can be expected for mixed duckweeds and the smallest for the presence of only one species of *L. minor*. Sediment increments in thickness were only observed until the end of the first month (Table 6). The largest area was covered at the highest rate by sediment formed by *L. trisulca*, reaching its maximum of 95% after two months. The highest dynamics of sediment coverage of the bottom was observed in the mixed duckweed variant, particularly under semi-shaded conditions of 1.53%. After around two months, however, the level stabilised and did not exceed 60% coverage under shaded conditions. By far, the smallest sediment cover was observed with *L. minor* stocking, which after stabilisation, after approximately 2 months, did not exceed 55% cover (Table 6).

## Nature of the relationship between O$_2$ and the ecological factors

Statistically significant correlations were found in trials with full light access, with an increased ambient temperature significantly reducing water oxygenation in all duckweed combinations. Lower values of water oxygen saturation were also influenced by the amount of light reaching the surface of the water surface, especially in single-species systems. Studies have shown that an increase in atmospheric pressure positively affects the duckweed's ability to produce oxygen, regardless of the *Lemna* combination. With both limited and full light, the temperature was not a determining factor for oxygen saturation in the water, and neither was light. With partial shading, however, the thickness of the plant layer developed just below the water surface, and the amount of deposited sediment was important (Fig. 6).

Among the analysed parameters, atmospheric pressure and water conductivity always showed significant effects on the amount of oxygen for all duckweed compositions. Table 7 presents significant linear regression models describing the dynamics of changes in dissolved oxygen as a function of these two relevant factors (irrespective of the duckweed type). In full light, oxygen variability was explained in a range from 26 to 42%, while in partial shade, it was explained in the 13–44% range. The remaining (unexplained) part of
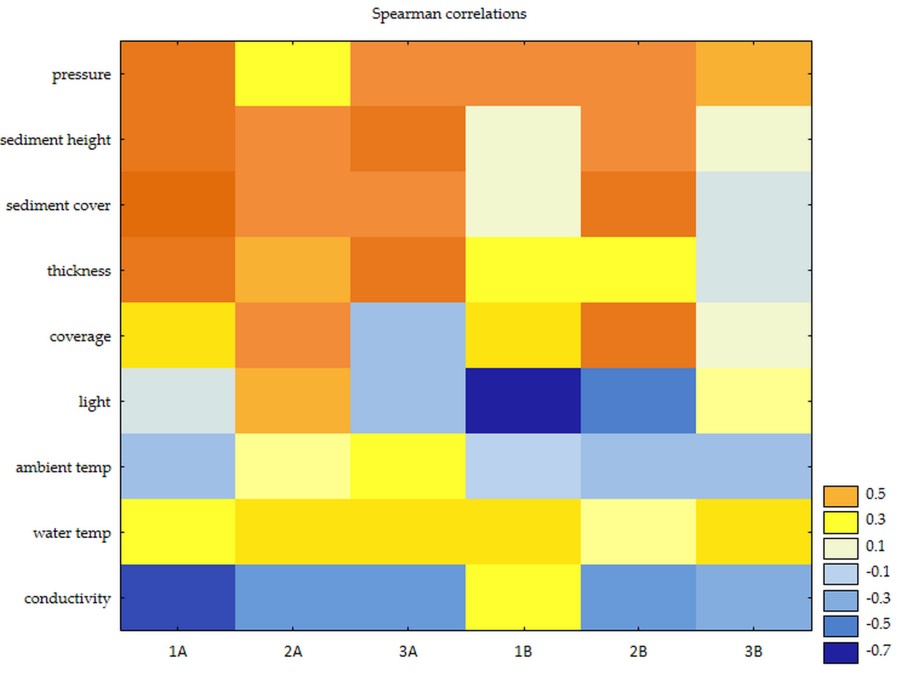

**Figure 6** Observed features in relation to water oxygen saturation (Spearman correlations): (A) shaded and (B) full light.

**Table 7  Dissolved oxygen change dynamics in water.**

| light | The dynamics of changes in dissolved oxygen in water | $R^2$ | F test (df: 1; 31) | *p*-value |
|---|---|---|---|---|
| Full | $O_2$ (%) = 0.30 pressure − 218.25 | 0.26 | 19.90 | 0.0001[*] |
| Full | $O_2$ (mg) = 0.03 pressure − 23.51 | 0.42 | 19.90 | 0.0001[*] |
| Full | $O_2$ (mg) = −0.003 conductivity + 8.297 | 0.36 | 19.90 | 0.0001[*] |
| Shade | $O_2$ (%) = 0.32 pressure − 242.68 | 0.17 | 19.90 | 0.0001[*] |
| Shade | $O_2$ (mg) = 0.02 pressure − 16.91 | 0.13 | 10.93 | 0.0024[*] |
| Shade | $O_2$ (mg) = −0.004 conductivity + 8.930 | 0.44 | 19.90 | 0.0001[*] |
| Shade | $O_2$ (%) = −0.042 conductivity + 103.383 | 0.33 | 12.95 | 0.0011[*] |

Notes.
[*]$p < 0.05$

the $O_2$ (%) and $O_2$ (mg/l) variability is a rationale for further exploration of factors that may be relevant to the process under study.

# DISCUSSION

## Shaping oxygen content concerning different compositions of duckweed species

Duckweeds are small, free-floating vascular plants that prefer nutrient-rich waters (*Lüönd, 1980*; *Yaseen & Scholz, 2016*). They have a great ability to take up and store many dissolved substances. They are therefore used in wastewater treatment, also in temperate climates (*Ozimek, 1998*; *Bekcan, Atar & Abdullah, 2009*). The use of native species in the treatment,

which the analysed duckweed species are for the European territory, is extremely important (*Wright & Jones, 2006*). The introduction of non-native species is ultimately not beneficial as confirmed by studies from shallow the tidal Hudson River (*Caraco et al., 2006*). Hence, it is important to rely on native species in modelling the flora of ecosystems. The most commonly used species in treatment plants in Poland is *L. minor*. A lot of information can be found on *L. minor*, while *L. trisulca* has been studied far less. According to *Keddy (1976)*, *L. minor* has greater colonisation abilities and is more competitive in relation to other species, as well as more resistant to drying. A water environment richer in nutrients, for example, fed with animal waste, can cause a rapid increase in *Lemna* biomass in a very short time (*Chepkirui et al., 2023*), which then absorbs more proteins and nitrogen (*Xu et al., 2011*). In oxbow lakes from which the plants originated, *L. minor* also dominated, mainly colonising the rushes zone. In contrast, *L. trisulca* was more common in the open, unshaded zone.

According to a study by *Pokorný & Rejmánková (1983)*, both dense and loose Lemnaceae coverages, consisting mainly of *L. minor* and *Spirodella polyrrhiza* (L.) Schleid. did not release oxygen into the medium, and oxygen uptake by bottom sediments reduced the oxygen concentration up to 2 mg/l. In support of our hypothesis that the quantity of oxygen varies between the two *Lemna* species and partial agreement with the research results of the cited authors, we found that the highest values of oxygen saturation of water occurred in the composition of mixed duckweeds with limited light access. In contrast, at full light access, they were colonised by *L. trisulca*. That species is not completely surface-associated species, as it develops just below the water surface, where the light is reflected (*Huebert, 1992*). Therefore, it can be concluded that *L. minor* releases more oxygen into the atmosphere, whereas *L. trisulca* releases it mainly to water, except during periods of water supersaturation (*Sculthorpe, 1967*). The high oxygen content found in the control samples is likely to be the result of oxygen diffusion from the environment (*Entradas, Waldron & Volk, 2020*; *Moriarty et al., 2017*).

## Factors influencing oxygen content in water

Among the analysed factors, atmospheric pressure was the parameter that had the most substantial effect on the amount of dissolved oxygen in water, regardless of the composition and amount of duckweed in the water. The amount of oxygen increased with increasing atmospheric pressure. According to Henry's law, this is a typically physical phenomenon (*Sander, 2015*). Both atmospheric pressure and conductivity appeared to be factors strongly related to the amount of oxygen in water. The conductivity of water is its ability to conduct electricity; pure water conducts electricity poorly. Conductivity due to organic impurities is also generally poor. The analysed samples ranged from 200 to 600 $\mu Scm^{-1}$, a value characteristic of eutrophic waters (*Choiński, 1995*). Conductivity provides information on the extent of mineralisation of waters, and therefore also on the level of pollution. Organic compounds found in water dissociate little or not at all. Dissolved organic compounds are most common in water (*Macioszczyk & Dobrzyński, 2002*). In our study, their source was plant decomposition. An increase in conductivity values significantly reduced the oxygen content of water, particularly in shaded samples. Solar energy is a source of both light and

heat. Their effects as ecological factors in the aquatic environment are different. Thermal radiation is the part of the solar spectrum with the longest wavelength (infrared radiation). Water temperature considerably impacts many phenomena and processes in ecosystems, including oxygenation. Thermal radiation raises water temperature, accelerating chemical reactions, including oxygen solubility in water.

An increase in temperature causes a decrease in the solubility of oxygen in the liquid, therefore lowering the rate of oxygen transfer into the liquid (*Truesdale, Downing & Lowden, 1955*; *Sand-Jensen & Pedersen, 2005*). Our study confirmed that an increase in ambient temperature significantly reduces water oxygenation in all duckweed combinations. Light is gradually absorbed in water. It is also strongly scattered and either converted to heat or other energy sources in photosynthesis, stored in organisms as reduced carbon. Plants only use part of the light spectrum. This is known as photosynthetically active radiation (*Zurzycki, 1970*; *Wilson & Meyers, 2007*). Research conducted by *Ojala & Julmala-Jäntti (2009)* indicates that *L. trisulca* tolerates shading better. Under full-light conditions, the amount of oxygen in the water in samples with *L. minor* and *L. trisulca* decreased significantly. At the same time, in semi-shade, it increased significantly, but only with *L. minor*. From a practical point of view, especially for hydrophytic treatment plants, it is advisable to create a structure properly, *e.g.*, reservoir banks, to ensure restrictions on full light access, which, on the one hand, raises the temperature and increases direct light access, and on the other hand reduces the oxygen content in water.

According to some authors, the presence of a dense cover formed by duckweed on the water surface inhibits oxygen entry into the water through diffusion from the air, as well as photosynthetic oxygen production by phytoplankton due to poor light penetration (*Brix & Schierup, 1990*; *Boedeltje et al., 2005*; *Ceschin, Crescenzi & Iannelli, 2020a*). However, *Lu et al. (2013)* showed that low-intensity shading by duckweed stimulates the efficiency of the photosynthetic process of submerged macrophytes rooted at the bottom of an eutrophic reservoir. *Zirschky & Reed (1988)* found that BOD (biological oxygen demand) can decrease in duckweed-covered ponds due to reduced oxygen transport to the water. In their study, cover was closely correlated with oxygen content in water only for *L. minor*. On the other hand, under light-limited conditions, the duckweed cover's thickness correlated significantly with the water's oxygen content. The higher it was, the higher the oxygen concentration in water. According to the study by *Körner, Lyatuu & Vermaat (1998)*, the presence of duckweed accelerates the decomposition processes of organic matter due to the additional supply of oxygen, despite the surface being wholly occupied by it. With full access to light, an increase in ambient temperature significantly reduces water oxygenation in all duckweed combinations. Lower oxygen saturation values were also due to more light reaching the water surface, particularly in single-species systems. *McIlraith (1988)* studied two species of *Lemna* that may compete for light and nutrients. One-way competition for light may give *L. minor* a competitive advantage in eutrophic habitats, which is also confirmed by our research. *Landesman, Fedler & Duan (2011)* state that duckweed populations can grow very densely in nutrient-rich environments. Layers of fronds grow one on top of another to form a mat that can reach a thickness of up to 6–10 cm (*Mkandawire et al., 2007*). This thick mat creates an anaerobic environment in

the water body on which the mat floats, therefore promoting anaerobic digestion and denitrification of the water body with duckweed. In our study, the maximum width of the plant for *L. trisulca* was 3.8 cm, while for *L. minor,* it was 1.45 cm. In such cases, anaerobic conditions did not occur, and the entire analysed water column was saturated with oxygen. Solar energy heats only the surface layer of water up to a depth of 1-2 m, whereas 30% of the heat is transferred. Water is a poor conductor of heat, hence, the movement of molecules in the liquid mass plays an important role. It is a substance with the highest specific heat. This property has a stabilising effect on thermal conditions (*Schwoerbel, 2016*). It can be expected that due to its heat storage capacity, the presence of duckweed, particularly *L. trisulca,* enhances the thermal stabilisation process in water. This, in turn, moderates the climate of adjacent areas, depending on the occupied water surface and duckweed cover. Our research showed that the presence of duckweed can retain additional heat reserves and, therefore, enhance the thermal stabilisation of water, but at the same time, potentially affect the rate of change in the water body itself. This issue may provide a basis for future research in this direction.

## The role of duckweed in detritus production

Duckweeds are aquatic macrophytes that have roots but grow unattached to the substratum. They show rapid growth, with a reported doubling time of 3 days, covering the entire water surface within 7∼10 days (*Landolt, 1986*; *Ziegler et al., 2015*; *Ceschin et al., 2016*). The mass of duckweed under average sunlight conditions in Poland doubles every 5-7 days, yielding up to 13 tons of dry matter per hectare annually (*Krzemieniewski, Zieliński & Debowski, 2007*; *Priya, Avishek & Pathak, 2012*). Generally, *Lemna* produce significant amounts of rapidly decomposing phytomass (detritus), containing 5–18 percent of cellulosic substances and 45-53 percent of biological substances (*Kathi, 2016*; *De Queiroz et al., 2020*). Their lifespan depends on temperature. The higher the temperature, the shorter the lifespan (*Sudiarto, Renggaman & Choi, 2019*; *Ceschin, Crescenzi & Iannelli, 2020a*). Lemnaceae live for an average of 5–7 weeks (*Cross, 2005*; *Szmeja, 2006*). In our study, under laboratory conditions, the peak growth of the analysed duckweed species was similar. The beginnings of morphological changes associated with chlorophyll loss and lack of growth were observed after about 20 days of culture. Under natural conditions, lifespan is defined as 3–6 weeks (*McIlraith, 1988*; *Lemon, Posluszny & Husband, 2001*). Flowering is a very rare phenomenon among these plants (*Lemon, Posluszny & Husband, 2001*), which was not observed also during the experiment. The greatest accumulation of dead organic matter occurred between days 30 and 40. From the point of view of the processes occurring in nature and, therefore, in *Lemna* hydrophytic treatment plants, it is interesting to see how the amount of phytomass produced by *Lemna* in the form of detritus changes over time. According to the arrangements of *Sumner, Amy & Talling (2008)*, the constant deposition of suspended solids at the bottom of the reservoir in the form of sediment reduces organic pollutants in the water column. Based on the above considerations, our study shows that the weakest purifier was *L. minor*. Sediment was formed most rapidly in the stocking with *L. trisulca*, while its greatest thickness was observed in the mixed duckweed variant, especially in the semi-shade. According to the predictive horizontal analyses carried out for sediment

accumulation under undisturbed (laboratory) conditions, we can predict how much , per year, sediment will increase with a certain coverage of the analysed species of duckweeds. In the case of the mixed *Lemna*, it will be about 5.3 cm per year, while for *L.trisulca* it will be 2 cm per year.

## CONCLUSIONS

Thanks to the study, the different ability of the analysed species of duckweeds to produce oxygen and organic matter was demonstrated, as well as the pointing out of the factors having the most significant impact on these processes. It turned out that the presence of *L. trisulca* causes higher dissolved oxygen content in water and the development of a large amount of dead organic matter. As a rule, full access to light in separate samples with *L. minor* and *L. trisulca* supported their growth while lowering the oxygen concentration. The mixed duckweed species simultaneously preferred light-restricted conditions. The lowest increase in the sediment layer was recorded for *L. minor*, which may indicate its smallest contribution to the purification of the water column and the shallowing of water reservoirs. Different light conditions, atmospheric pressure and conductivity only partially explained the variability of oxygen production by the tested duckweed combinations. This constitutes a premise for further search for essential factors for the examined process. Certainly, a major contribution of the conducted studies has been the analysis of the amount of organic matter (detritus) produced by *Lemna*, which were not analysed in detail. Particularly valuable seems to be the information on *L. trisulca*, which was not a frequent object of study, and whose contribution to both oxygen and organic matter production was greater than *L. minor*. The continuation of the study seems necessary, because complete knowledge will be gained only from studies conducted under outdoor conditions, with consideration of varying trophic levels, seasonal variation and different management of the surrounding area. Knowledge about which species of duckweed can serve as better oxygen transfers and producers of organic matter, under specific thermal and light conditions, allows us to extrapolate these values to facilities requiring human intervention (wastewater treatment plants, ponds, retention reservoirs), where the composition of pleustophytes can be shaped.

### Funding
The authors received no funding for this work.

### Competing Interests
The authors declare there are no competing interests.

### Author Contributions
- Joanna Sender conceived and designed the experiments, performed the experiments, prepared figures and/or tables, authored or reviewed drafts of the article, and approved the final draft.

- Monika Różańska-Boczula analyzed the data, prepared figures and/or tables, authored or reviewed drafts of the article, and approved the final draft.

## Data Availability

The raw measurements are available in the Supplementary File: Raw data.xlsx. The raw data present the results of the pilot (preliminary) study and the main experiment.

For each Lemna spp. studied, a specific set of traits was measured to verify environmental relationships.

## Supplemental Information

Supplemental information for this article can be found online at http://dx.doi.org/10.7717/peerj.17322#supplemental-information.

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
