# Peer review of "Preliminary studies of selected Lemna species on the oxygen production potential in relation to some ecological factors"

_PeerJ, doi:10.7717/peerj.17322_

## Round 0.1 · original submission · Major Revisions

The greatest weakness in this study is that the findings are preliminary and the study has not been placed in a wider context with identified knowledge gaps and contributions to the existing body of knowledge. As a result, the manuscript should be improved at all stages from the introduction, descriptions of the methods, presentation of the results, discussion and conclusions should be improved as indicated by the reviewers. The manuscript should be significantly improved for it to be considered for publication in PeerJ.

**Language Note:** The review process has identified that the English language must be improved. PeerJ can provide language editing services - please contact us at [email protected] for pricing (be sure to provide your manuscript number and title). Alternatively, you should make your own arrangements to improve the language quality and provide details in your response letter. – PeerJ Staff

·

Basic reporting

Abstract
 Results not well presented
 conclusions and recommendations are missing
Introduction
 Line 36- They among others….. which others? Specify and consider paraphrasing that statement
 Line 38, 42- O2 instead of O2
 Line 43-clearly describe DO distribution in water profile
 Line 51, 59, 69, 79-Avoid very old citation e.g. Kemp et al., 1984, Podbielkowski and 60 Tomaszewicz, 1996, Ozimek, 1991, Rahmani and Sternberg, 1999 etc
 Line 61- why duckweed are valuable food? Describe its nutritional composition
 Line 64- give reasons why Lemna minor and Lemna trisulca are the most dominant in Poland
 Line 78- Ease of culture…. Cite the latest publications and avoid old publications e.g. Chepkirui et al., 2023, Chakrabarti et al., 2018
Overall comments: Introduction
 Give general description of duckweed distribution in a funnel overview
 Clearly describe the morphology of duckweed
 Avoid citing oldest publications, consider current manuscripts
 Consider structuring the paragraphs well
 Alot of grammatical and tenses errors e.g. future tense instead of past tense
 Insufficient background information on duckweed

Experimental design

Gives us the coordinates and average depths of oxbow lakes in the Nadwiepvzanski Landscape Park
 Both species occurred in the same habitat, how did you separate the two?
 Line 110- what did you use to sterilized duckweed?
 Line 113- Duckweed were cultivated on nutrient media, what type of nutrients did you used? Fertilization rate? Water level/height?
 How much biomass (g/%) did you introduced into the culture unit?
 Line 117- why 2 controls and not 4?
 Data collection procedures missing. For instance, how did you collect data on duckweed height, amount, thickness and coverage of sediments.
Overall comment: Methodology not well structured

Validity of the findings

Line 172- what is the p value?
 Line 176- slightly higher by what percentage?
 Line 180-Give values for oxygen concentration, not just reporting as higher or lower. The same also applies to other parameters
 Line 185- paraphrase your sentences
 Line 203- The mean value of water oxygen saturation in the different duckweed systems varied, with the highest values in samples from Lemna trisulca in full light? Give us the values, what was the highest values?
 Line 207-Give treatments where specific significance difference was observed according to post doc analysis
 Line 209, 210, 251, 256- mix up of results and discussion
 Line 255- what do you mean by fine duckweed?
 Line 257- …. In the first several days…. Mention the specific days.
 Line 264- give specific strain with high sediment cover
 Line 291-There was also a clear relationship in the case of atmospheric pressure an increase in which contributed to an increase in oxygenation values. Paraphrase this statement.
 Consider summarizing table 4,5,6 under one table rather than having so many tables
Overall comment: Results not well presented, consider specifying values not just highest, lowest, there was or no significant difference etc. Avoid discussion under results section.

Additional comments

Discussion
 Line 316- all scientific names should be italicized
 Line 319- give reasons behind high colonization of L.triscula in open, unshaded zones
 Avoid old citations
 Avoid bringing results again in discussion section, a lot of repetitions
 Line 378- apart from width, was there any difference in the root length between the two species?
 Line 399- Does it depend on temperature alone? What of nutrients availability?
 Line 400- cross check the duckweed’s lifespan from the literature
Overall comment: No comparison between the present results and the past studies
Conclusions
 Give strong conclusion and summarize inform of paragraphs not point form
Recommendations
 This section is missing
 Give gaps for further study

Reviewer 2 ·

Basic reporting

no comment

Experimental design

no comment

Validity of the findings

no comment

Additional comments

The paper entitled " Effect of selected Lemna species on dissolved oxygen content
in water" emphasized on the effect of different duckweed species (Lemna minor and Lemna trisulca) on dissolved oxygen content in water.
However, this manuscript is a simple and preliminary study and the result did not fully support the conclusion. Thus, I think it is not suitable for publication in Journal of Peer J.

Reviewer 3 ·

Basic reporting

The article is written clearly and in correct English.

The article include sufficient introduction and background to demonstrate how the work fits into the broader field of knowledge. Relevant prior literature should be improved by adding other appropriated references and more recent

The structure of the article is conform to a starndard format.However, too many figures and tables have been inserted. It is suggested to merge some tables (see attached file).

The submission is "self-contained,’ and represent an appropriate ‘unit of publication’. It includes all results relevant to the hypothesis.

Experimental design

Original primary research within Aims and Scope of the journal. YES

Research question well enough defined, relevant & meaningful. It is stated how research fills an identified knowledge gap. The aims should be written clearer

The investigation has been conducted rigorously and to a moderate technical standard. The research has been conducted in conformity with the prevailing ethical standards in the field.

Methods should be described with more information and clearer (see attached file)

Validity of the findings

Data have been provided correctly; they are good and statistically sound.
Conclusions should be improved without repeating the main obtained results.

Additional comments

Comments and suggestions of corrections and changes are reported directely in the attached manuscript file.

Annotated reviews are not available for download in order to protect the identity of reviewers who chose to remain anonymous.

---

## Round 0.2 · Major Revisions

I now review the comments of one reviewer who evaluated the previous version of the manuscripts. There reviewer is not satisfied with the quality of your manuscript and has made further comments and recommendations to improve your work. I have also read your manuscript and noted several issues that you must address. Although not comprehensive, my review noted that you need to improve the clarity of your writing and there are far too many typos that should not be in a manuscript at this stage of revision. Carefully go through your manuscript and improve the grammar and clarity of statements. The formatting also needs to be improved.

Below are some of my comments:
Ln 20: Delete ‘in water’
Ln 23: Delete the reference
Ln 24: free-floating vascular plants
Ln 24-25: What is being compared here? The statement is not clear.
Ln 27: Dense mats!
Ln 20-51: Write the abstract in one paragraph and observe the maximum limit of 500 words.
Ln 56: Delete the full stop after organisms
Ln 77-80: You can’t have a paragraph of two sentences!
Ln 94: Provide a reference for these numbers
Ln 111: ‘The severely reduces’- correct for clarity
Ln 134-138: This is a very long and complex sentence. Break for clarity.
Ln 156: ‘Official Journal of the European Union 2009’ – cite this correctly
Ln 163: ‘a place with no regulated varying temperature’ – this is not clear
Ln 221-229: report actual numbers in Mg/L too, not percentages. If you must, this can be additional.

The most important question is how your controlled experiment can be scaled to natural conditions. You need to address this question in your discussion and conclusions

·

Basic reporting

Title: Not clear, need further structuring
Paragraphs not flowing with some repetitions e.g. Line 81 & 91. Consider rearranging.
Hanging statements e.g. Line 121, 400, 430 etc.

Experimental design

Experimental protocols for in-situ plant collection are missing.
Line 148-preliminary analysis done missing.

Validity of the findings

consider comparing the current findings with the previous studies.

Additional comments

Choose whether to include DOI or not in reference section for purposes of consistency throughout the text.

---

## Round 0.3 · accepted · Accept

The authors have made all the suggested revisions, and the manuscript is now much improved and acceptable for publication.

Reviewer 3 ·

Basic reporting

The authors greatly improved the paper in this second version. All the changes that were suggested were made by the authors.

Experimental design

no comment

Validity of the findings

The authors greatly improved the paper in this second version.

Additional comments

The authors greatly improved the paper in this second version. All the changes that were suggested were made by the authors.